# Carbonate Apatite and Hydroxyapatite Formulated with Minimal Ingredients to Deliver SiRNA into Breast Cancer Cells In Vitro and In Vivo

**DOI:** 10.3390/jfb11030063

**Published:** 2020-09-10

**Authors:** Rowshan Ara Islam, Hamed Al-Busaidi, Rahela Zaman, Syafiq Asnawi Zainal Abidin, Iekhsan Othman, Ezharul Hoque Chowdhury

**Affiliations:** 1Jeffrey Cheah School of Medicine and Health Sciences, Monash University Malaysia, Jalan Lagoon Selatan, Bandar Sunway, Subang Jaya 47500, Selangor, Malaysia; Rowshan.Islam@monash.edu (R.A.I.); hamed.al-busaidi@monash.edu (H.A.-B.); rahela.zaman@monash.edu (R.Z.); syafiqnawi@gmail.com (S.A.Z.A.); iekhsan.othman@monash.edu (I.O.); 2Health and Wellbeing Cluster, Global Asia in the 21st Century (GA21) Platform, Monash University Malaysia, Jalan Lagoon Selatan, Bandar Sunway, Subang Jaya 47500, Selangor, Malaysia

**Keywords:** carbonate apatite, hydroxyapatite, inorganic nanoparticles, breast cancer, protein corona, keratin

## Abstract

Introduction: Cancer is one of the top-ranked noncommunicable diseases causing deaths to nine million people and affecting almost double worldwide in 2018. Tremendous advancement in surgery, chemotherapy, radiation and targeted immunotherapy have improved the rate of cure and disease-free survival. As genetic mutations vary in different cancers, potential of customized treatment to silence the problem gene/s at the translational level is being explored too. Yet delivering therapeutics at the required dosage only to the affected cells without affecting the healthy ones, is a big hurdle to be overcome. Scientists worldwide have been working to invent a smart drug delivery system for targeted delivery of therapeutics to tumor tissues only. As part of such an effort, few organic nanocarriers went to clinical trials, while inorganic nanoparticles (NPs) are still in development stage despite their many customizable properties. Carbonate apatite (CA), a pH sensitive nanocarrier has emerged as an efficient delivery system for drugs, plasmids and siRNAs in preclinical models of breast and colon cancers. Like hydroxyapatite (HA) which serves as a classical tool for delivery of genetic materials such as siRNA and plasmid, CA is an apatite-based synthetic carrier. We developed simplified methods of formulating CA-in-DMEM and a DMEM-mimicking buffer and HA in a HEPES-buffered solution and characterized them in terms of size, stability, protein corona (PC) composition, cytotoxicity, siRNA delivery efficiency in breast cancer cells and siRNA biodistribution profile in a mouse model of breast cancer. Methods: Particle growth was analyzed via spectrophotometry and light microscopy, size was measured via dynamic light scattering and scanning electron microscopy and confirmation of functional groups in apatite structures was made by FT-IR. siRNA-binding was analyzed via spectrophotometry. Stability of the formulation solutions/buffers was tested over various time points and at different temperatures to determine their compatibility in the context of practical usage. Cellular uptake was studied via fluorescence microscopy. MTT assay was performed to measure the cytotoxicity of the NPs. Liquid chromatography—mass spectrometry was carried out to analyze the PC formed around all three different NPs in serum-containing media. To explore biodistribution of all the formulations, fluorescence-labeled siRNA-loaded NPs were administered intravenously prior to analysis of fluorescence intensity in the collected organs and tumors of the treated mice. Results: The size of NPs in 10% serum-containing media was dramatically different where CA-in-DMB and HA were much larger than CA-in-DMEM. Effect of media was notable on the PC composition of all three NPs. All three NPs bound albumin and some common protease inhibitors involved in bone metabolism due to their compositional similarity to our bone materials. Moreover, CA also bound heme-binding proteins and opsonins. Unlike CA, HA bound different kinds of keratins. Difference in PC constitution was likely to influence accumulation of NPs in various organs including those of reticuloendothelial system, such as liver and spleen and the tumor. We found 10 times more tumor accumulation of CA-in-DMB than CA-in-DMEM, which could be due to more stable siRNA-binding and distinct PC composition of the former. Conclusion: As a nanocarrier CA is more efficient than HA for siRNA delivery to the tumor. CA prepared in a buffer containing only the mere constituents was potentially more efficient than classical CA prepared in DMEM, owing to the exclusion of interference attributed by the inorganic ions and organic molecules present in DMEM.

## 1. Introduction

Cancer is one of the leading fatal diseases worldwide. Along with surgery, other major parts of standard cancer treatment include chemotherapy which kills mitotic cells indiscriminately, hormonal therapy and receptor targeted therapy which are more specific to cancer cells. Cancer is the result of genetic mutations of oncogenes or tumor suppressor genes that vary among individuals. With the help of microarray and proteomics analysis, it is possible to determine the faulty genes in an individual and suggest a very personalized treatment regimen to subside overexpression of defective genes at mRNA level with siRNAs [1]. Chemical modification of siRNA enables its systemic delivery in free form, avoiding its rapid degradation by serum nucleases and clearance by hepatic and renal systems [2]. However, modification may compromise the silencing efficiency and produce toxic metabolites, demonstrating a need for carriers which can selectively deliver siRNA to target tissues without off-loading the cargo elsewhere and can escape endosomal degradation once inside the cell [3]. Viral vectors are amazingly efficient; however, the risks of immunogenicity and carcinogenicity limit their broad usage [4]. Also, viral vectors are not suitable for delivery of siRNA which does not need to be transcribed, like shRNA. Among the organic nanocarriers, a few liposomal formulations carrying specific siRNAs, such as ALN-VSP 02 and Atu027-I-02 have reached Phase I clinical trial for cancer treatment, while a Phase I trial for CALAA-01, a cyclodextrin polymer containing a siRNA against M2 subunit of ribonucleotide reductase (R2) was terminated [5,6]. An issue with organic nanocarriers is their hectic preparation process [7]. Compared to viral and organic nanocarriers, the field of inorganic nanocarriers for siRNA delivery is currently at an early stage of development for clinical applications. A few widely studied inorganic NPs are gold (Au) NPs, magnetic NPs, carbon nanotube, graphene, quantum dot, etc [8]. The tunable properties of inorganic NPs such as size, shape, customizable surface features, PC, and loading efficiency have made them attractive options for the delivery of genetic materials [9]. However, inorganic nanocarriers for the delivery of genetic materials are still in early stage of development. Optimization of transfection efficiency while minimizing cytotoxicity of inorganic NPs still requires further research to proceed from bench to bedside.

HA, Ca_10_(PO_4_)_6_(OH)_2_, has been in the race of promising gene carriers since long; however its heterogeneity, aggregation and unhandy preparation process are still the barriers to be overcome [10,11]. CA is a recently developed inorganic nanocarrier basically comprising Ca^2+^, inorganic phosphate and carbonate ions. In terms of chemical composition, this is a derivative of HA formed by substituting some phosphate ions with carbonate ions with a final molecular formula Ca_10_(PO_4_)_6−X_(CO_3_)_X_(OH)_2_. The easy fabrication method takes only 30 min at 37 °C. CA NPs have been used successfully in our lab for in vitro and in vivo delivery of anticancer drugs, such as taxanes (paclitaxel and docetaxel), doxorubicin, cisplatin, gemcitabine, siRNAs against genes of growth factors, antiapoptotic proteins, cell adhesion molecules and plasmids carrying genes p21 and p53 in an animal model of breast cancer [12,13,14,15,16,17,18,19,20]. Upon cellular internalization by endocytosis, particles are thought to be rapidly dissolved into constituent ions such as Ca^2+^, PO_4_^3−^ and HCO_3_^−^ in endosomal acidic pH, leading to an increase in osmotic pressure across the endosomal membrane, consequential water entry into the endosome, swelling and rupturing of the endosome, and ultimately releasing the siRNA, plasmid or drug in the cytoplasm.

HA and derivatives are advantageous over many other NPs for their biocompatibility and less toxicity [21]. Both HA and CA can be subjected to surface modification, which can remarkably improve their cellular uptake and delivery efficiency [17,22]. The surface properties, such as charge and coating can determine its interaction with proteins in systemic circulation, biodistribution, retention and clearance from the body [23]. In Fact, PC may play a prominent role in exploiting the enhanced permeation and retention (EPR) effect in solid tumors. Discovered by Matsumura and Meda in 1986, EPR effect explains that NPs can take advantage of the poor vasculature of tumor tissues with wide fenestrations and inefficient lymphatic drainage to accumulate in tumor tissues [24,25].

In this article, we have carried out a comparative study on physicochemical properties, PC composition and siRNA delivery efficacy of classical CA prepared in bicarbonated DMEM, CA prepared with the mere constituents, namely calcium, phosphate and bicarbonate, in an aqueous buffer which we referred as DMEM-mimicking buffer (DMB) and HA prepared in a simplified manner in a HEPES-buffered solution (HBS) of pH 7.5. All three inorganic NPs were characterized in terms of size, shape, cellular uptake, efficiency to deliver siRNA in vitro and in vivo and PC formed in presence of blood plasma. The time- and temperature-dependent stability of all the buffers were also reported.

ATR-FTIR analysis confirmed the presence of functional groups for valid CA and HA composition. All three differently formulated NPs showed an increasing trend of growth with increasing Ca^2+^ concentration in terms of particle number and size. These pH-sensitive NPs were capable of loading a significant percentage of siRNA. Constituents in DMEM played regulatory roles to limit particle size—and when integrated in the NPs—caused loose siRNA-binding. DMEM and HBS buffers showed stability, which is an important feature for their practical usage at different temperatures and periods. In contrast, DMB was not stable over longer durations. As a vehicle, differently formulated CAs were not cytotoxic, while HA was toxic to the cells in vitro. Among the three NPs, CA showed better siRNA delivery to the cells in vitro and to the tumors in vivo. CA prepared in DMB showed 10 times more accumulation of siRNA in tumor tissue compared to classical CA prepared in DMEM. We did not find an accumulation of siRNA in tumors after four hours of systemic delivery of siRNA loaded HA. From the biodistribution study, it was clear that reticuloendothelial system was the major route of excretion for all three NPs, which can be explained by the presence of fetuin protease inhibitors in the PC. PC composition predictably also influenced cellular uptake and biodistribution of these NPs. From this study, we conclude that CA was better than HA as an siRNA delivery system. Between the two differently formulated CAs, CA-in-DMB showed higher siRNA delivery to the tumors.

## 2. Methods and Materials

### 2.1. Reagents

Dulbecco’s modified Eagle’s medium (DMEM), calcium chloride dihydrate (CaCl_2_·2H_2_O), sodium bicarbonate (NaHCO_3_), sodium phosphate monobasic monohydrate (NaH_2_PO·H_2_O), sodium chloride (NaCl), glucose, HEPES, dimethyl sulfoxide (DMSO) and thiazolyl blue tetrazolium bromide (MTT) were purchased from Sigma-Aldrich (St Louis, MO, USA). DMEM powder, fetal bovine serum (FBS), trypsin-ethylenediaminetetraacetate (trypsin-EDTA) and penicillin–streptomycin were obtained from Gibco BRL (Carlsbad, CA, USA). MS Grade Pierce Trypsin Protease was obtained from Thermo Fisher Scientific (Waltham, MA, USA). Allstars negative siRNA AF 488 (5 nmol) was purchased from QIAGEN. MCF7 and 4T1 cells were obtained from ATCC. Six to eight weeks old BALB/c female mice were bred and maintained at the Animal Facility Unit in Monash University (Sunway campus, Malaysia).

### 2.2. Preparation of NPs

CA was prepared in DMEM or DMB. One hundred milliliters of DMEM was prepared with 1.35 g of DMEM powder and 0.37 g of sodium bicarbonate. DMB was prepared by adding 110.34-mM sodium chloride, 0.906-mM sodium phosphate monobasic monohydrate, 44-mM sodium bicarbonate and 25-mM glucose in water. The concentrations of the constituents were same as that in DMEM powder (Thermo Fisher Scientific, Waltham, MA, USA, catalog no.12800). HA was prepared in HEPES-buffered solution (HBS). HBS was prepared with 25-mM HEPES, 110.34-mM sodium chloride, 0.906-mM sodium phosphate monobasic monohydrate and 25-mM glucose in water. The pH was adjusted to 7.5 for DMEM, DMB and HBS. To prepare NPs, different amounts of 1-M calcium chloride dihydrate were then added into the microcentrifuge tubes, followed by addition of DMEM/DMB/HBS up to 1 mL and incubation at 37 °C for 30 min. After incubation, 10% FBS was added to each microcentrifuge tube.

### 2.3. Assessment of Particle Growth Via Turbidity Measurement

CA particles were prepared in DMEM as described in Section 2.2 with 0 to 8 mM of CaCl_2_·2H_2_O. To determine the optimum Ca^2+^ concentration, CA particles in DMB and HA particles were prepared with a wide range of Ca^2+^, 0 to 175 mM. After incubation, absorbance was immediately measured at 320 nm using a UV spectrophotometer (V-630 UV-vis, JASCO, Easton, MD, USA).

### 2.4. Microscopic Observation of NPs

NPs were prepared as described in Section 2.2. After incubation samples were immediately transferred to a 24-well plate, left to settle down for 5 min before observation under a microscope (Olympus IX81 motorized inverted research microscope, Tokyo, Japan).

### 2.5. Size Measurement by Dynamic Light Scattering

NPs were prepared as stated in Section 2.2. After incubation, 10% FBS was added to each tube. Size was measured by Malvern Nano Zetasizer (Worcestershire, UK) and accompanying software. Samples were prepared and measured in duplicates and an average value (±S.D.) was calculated.

### 2.6. Size Measurement with Field Emission Scanning Electron Microscope (FE-SEM)

NPs were prepared in 1 mL volume in a 1.5-mL microcentrifuge tube with 6-mM and 10-mM Ca^2+^ for CA and HA, respectively. After incubation, the samples were centrifuged at 13,000 rpm for 15 min. After the supernatant was discarded, 1 mL ultrapure water was added, and samples were centrifuged again at the same condition. Carefully, 975 μL supernatant was removed by micropipette. The pellet was dissolved in 25 μL of the leftover water from which a 3-μL sample was transferred on a cover slip and dried at 37 °C for 1 h. The dried sample was subjected to platinum sputtering for 45 s at a current of 30 mA and a tooling factor of 2.3. The particles were visualized at 5 kV with a scanning electron microscope (Hitachi SU8010, Tokyo, Japan).

### 2.7. Attenuated Total Reflection-Fourier Transform Infrared Spectroscopy (ATR-FTIR)

NPs were prepared in 100 mL DMEM, DMB or HBS. Samples were centrifuged at 3000 rpm and at 4 °C for 45 min. Supernatant was replaced with water before re-centrifugation at the same condition. After centrifugation supernatant was removed leaving very little amount of water with pellet. Pellet was mixed with leftover supernatant and stored at 4 °C overnight. The next day samples were dried using a freeze dryer (Labconco, Kansas city, MO, USA). After 3 h, the powdered samples were taken out for analysis. Attenuated total reflectance sampling technique was used in a Varian FTIR machine (Santa Clara, CA, USA). Varian Resolutions Pro 640 software was used to analyze the spectra.

### 2.8. pH Sensitivity of the NPs

CA-in-DMEM, CA-in-DMB and HA were prepared with 4 µL, 6 µL and 10 µL from a 1-M stock of CaCl_2_·2H_2_O in 200 μL volume. After incubation, 800 μL DMEM, DMB or HBS of pH 7.5, 7.0 or 6.5-was added to the tubes. After 5 min, absorbance was measured at 320 nm.

### 2.9. Measurement of siRNA-Binding Efficiency

A Standard curve was plotted for different concentrations of fluorescent siRNA in 200 µL of 10 mM EDTA in water. Fluorescence intensity was measured at an excitation wavelength of 490 nm and an emission wavelength of 535 nm in a VICTOR™ X5 multilabel plate reader (Perkin Elmer, Waltham, MA, USA). CA-in-DMEM, CA-in-DMB and HA were prepared in 1-mL volumes with 4 mM, 6 mM and 10 mM of exogenous Ca^2+^ and 1, 2 or 10 nM of fluorescent siRNA, respectively. The particles were centrifuged at 13,200 rpm for 15 min at 4 °C. Afterwards, the supernatant was removed completely, and the pellet was dissolved in 200 µL of 10-mM EDTA, for which fluorescence intensity was measured. In a separate experiment, the pellet was washed with water by centrifuging at 13,200 rpm for 15 min at 4 °C, followed by dissolution of the pellet in EDTA before fluorescence measurement. The binding affinity of siRNA to the NPs was calculated using the following formula:(1)$$\%\;\mathrm{of}\;\mathrm{siRNA}\;\mathrm{binding}=\frac{X\left(\mathrm{pellet}\right)}{\;X\left(\mathrm{initial}\right)}\;\times 100$$
where X(pellet) is the concentrations of siRNA bound to NPs dissolved in EDTA and X(initial) denotes the total concentration of siRNA used in the experiment, which was 10 nM. The experiment was done in triplicates and data were reported as mean ± S.D..

### 2.10. Stability Analysis for NPs

DMEM, DMB and HBS were prepared in 500 mL volume and stored in 50-mL aliquots at different temperatures, +4 °C, −20 °C, −80 °C. After 1 day, 7 days or 30 days the buffers were taken out to prepare NPs with. Absorbance was measured at 320 nm. In a separate experiment, CA-in-DMEM was prepared and stored in different temperatures—+4 °C, −20 °C, −80 °C. Samples were taken out and brought to room temperature before measuring absorbance at 320-nm wavelength.

### 2.11. Cell Culture and Seeding

Human MCF-7 and mouse 4T1 cells were grown in T25 culture flasks in DMEM supplemented with 10% FBS in a humidified atmosphere having 5% CO_2_ at 37 °C. Exponentially growing cells were trypsinized and following an addition of fresh medium, the cell suspension was centrifuged at 120 rcf for 5 min and the supernatant was discarded. Fresh medium was added to resuspend the pellet; cells were counted using hemocytometer.

For seeding, an appropriate dilution was made using culture medium to produce a cell suspension of desired concentration. One milliliter of the prepared cell suspension was subsequently added into each well of a 24-well plate and allowed to attach overnight at 37 °C and 5% CO_2_. The next day, cells were treated with NPs followed by MTT assay after 48 h.

### 2.12. Cell Viability Assessment with 3-(4,5-Dimethylthiazol-2-yl)-2,5 Diphenyltetrazolium Bromide (MTT) Assay

Following 44 h of treatment with NPs, the fraction of viable cells was determined using MTT assay. Briefly, 50 μL of MTT (5-mg/mL in PBS) was added aseptically into each well, followed by incubation at 37 °C for 4 h. After the incubation, medium containing MTT was aspirated and the purple formazan crystals at the bottom of each well were dissolved in 300 μL of DMSO solution. The absorbance of the resulting formazan solution was then determined at a wavelength of 595 nm with a reference at 630 nm using microplate reader spectrophotometer (Bio-Rad Benchmark Plus, Hercules, CA, USA). Each experiment was performed in triplicates and the data were plotted as mean ± standard deviation (S.D.) of three replicates. The following are the formulas used to calculate viability and cytotoxicity:(2)$$\%\;\mathrm{Cell}\;\mathrm{Viability}=\frac{\mathrm{Absorbance}\;\mathrm{of}\;\mathrm{treated}\;\mathrm{cells}}{\mathrm{Absorbance}\;\mathrm{of}\;\mathrm{untreated}\;\mathrm{cells}}\;\times \;100$$
(3) % Cell Death =100−% Cell Viability 
(4) Actual Cytotoxicity of siRNA =Cytotoxicity of NP bound siRNA−Cytotoxicity of NP 

### 2.13. Microscopic Observation of Cellular Uptake of NPs Bound Fluorescent SiRNA

CA-in-DMEM, CA-in-DMB and HA were prepared with 10-nM AllStars Neg. siRNA AF 488. As CA-in-DMB and HA were prepared in water-based buffer that lacked the nutrients required for cell growth, the prepared NPs were diluted ten times in serum-containing medium before adding to cells. For this experiment, CA-in-DMEM was diluted ten times as a control. After 4 h of treatment, the media was removed and cells were washed with 100 μL of 5-mM EDTA in PBS to remove any unbound NPs from cell surface, followed by 2× wash with PBS. Cells were submerged in 100 μL PBS and viewed under Olympus IX81 motorized inverted microscope.

### 2.14. Protein Corona Identification via LC–MS/MS

#### 2.14.1. Sample Preparation for Protein Corona (PC) Analysis

For the PC analysis, NPs were prepared as stated in Section 2.2, with 6 mM and 10 mM of total (endogenous + exogenous) Ca^2+^ for CA and HA, respectively, followed by a 30-min incubation at 37 °C with mouse plasma. The sample was centrifuged at 13,200 rpm at 20 °C for 15 min. The supernatant was then removed and replaced with 1 mL water, without disturbing the pellet. The sample was centrifuged a second time under the same conditions, and the supernatant was discarded. The precipitate was resuspended in 100 μL of 50-mM EDTA, followed by desalting with C18 spin column (Thermo Fisher Scientific, Waltham, MA, USA). Solvents were evaporated in a centrifugal evaporator (EYELA Centrifugal Evaporator CVE 3110, Tokyo, Japan). Proteins were subjected to trypsin digestion.

#### 2.14.2. In-Solution Tryptic Digestion of Protein

The pellet formed from the evaporated sample was resuspended in 100 μL of 0.1% formic acid. The proteins were denatured and reduced by incubating at 60 °C for 60 min with 25 μL of 100-mM ammonium bicarbonate, 25 μL TFE (2,2,2-trifluoroethanol) and 1 μL of 200-mM DTT (dithiothreitol). After this, to alkylate the sample, 4 μL of 200-mM iodoacetamide (IAM) was added and incubated in the dark at room temperature. After 60 min, 1 μL of 200-mM DTT was added to destroy excess IAM and incubated at room temperature in the dark for 60 min. Next, 400 μL of 25-mM ammonium bicarbonate was added to raise the pH to 7–9, followed by addition of trypsin and incubation at 37 °C for 18 h. After digestion, 1 μL of TFA was added to lower the pH and thus to stop trypsin activity. The sample was dried in a vacuum concentrator.

#### 2.14.3. Sample Preparation for Q-TOF Mass Spectrometry

Thirty microliters of 0.1% formic acid in water was added to the dry digest before the mixture was vortexed briefly and sonicated in water bath for 5 min. Ice was added to keep the samples cool during sonication. Afterwards, the samples were centrifuged at 13,200 g for 10 min. One microliter of supernatant was injected into the MS Tube, which was subsequently placed on an autosampler before starting the LC-QTOF analysis.

#### 2.14.4. Nanoflow Liquid Chromatography Electrospray—Ionization Coupled with LC–MS/MS

The digested peptides were loaded into an Agilent C18 300 Å large capacity chip column that was equilibrated with 0.1% formic acid in water (solution A). The peptides were eluted from the column with 90% acetonitrile in water with 0.1% formic acid (solution B) using the gradients of 5% solution B over 0–45 min and 70% solution B over 45–55 min. The Q-TOF polarity was set at positive with capillary and fragmentor voltage being set at 1900 V and 360 V, respectively and 5 L/min of gas flow with a temperature of 325 °C. The peptide spectrum was analyzed in auto MS mode ranging from 110–3000 m/z for MS scan and 50–3000 m/z for MS/MS scan. Acquisition rates were 2 (spectra/s) for MS and 4 (spectra/s) for MS/MS. The spectrum was then analyzed with Agilent MassHunter (Agilent Technologies, Santa Clara, CA, USA) data acquisition software and PEAKS 8.0 software (Bioinformatics Solutions, Inc., Waterloo, ON, Canada).

#### 2.14.5. Protein Identification by Automated De Novo Sequencing (PEAKS Studio 8.0)

Protein identification was performed with PEAKS Studio 8.0 (Bioinformatics Solution, Inc., Waterloo, ON, Canada). Uniprot *Mus musculus* (Mouse) database (April 2018) was used for protein identification and homology search by comparing the de novo sequence tag. Carbamidomethylation was set as the fixed modification with maximum mixed cleavages at 3. Parent mass and fragment mass error tolerance were both set at 0.1 Da with monoisotopic mass as the precursor mass search type. Trypsin was selected as the enzyme used for digestion. False discovery rate of 1% and unique peptides 1 were used for filtering out inaccurate proteins. Only proteins showing high confidence levels (−10lgP > 15) in PEAKS were chosen, as it targets very few decoy matches above that threshold.

### 2.15. Biodistribution of NPs in a Mouse Model of Breast Cancer

The Monash University Animal Ethics Committee approved all the procedures involved in this experiment (MARP/2016/126 A). A total of 5 × 10^5^ 4T1 cells in 100 μL PBS were injected in the mammary fat pad of a mouse to induce an orthotopic breast tumor (considered as Day 1) and the mice were checked regularly for the outgrowth of tumor by touching the area of injection. On Day 12, when tumors were 105 ± 8 mm^3^ in volume, mice were administered with fluorescent AF-488-labeled neg. siRNA either free or bound to any of the NPs: CA-in-DMEM, CA-in-DMB and HA. Mice were euthanized in a CO2 chamber after 4 h of intravenous injection. After sacrificing, the liver, spleen, kidney, lung, heart, brain and tumor were collected, weighed and dipped in 500-μL lysis buffer (0.072 gm DTT, 1% NP-40 in 50 mL of 1X PBS) and stored in −80 °C until processing. Organs were chopped using sharp blade and crushed using mortar and pestle. The crushed tissues in lysis buffer were centrifuged at 8000 rpm for 20 min. Then, 150 μL of supernatant from each sample was transferred to 96-well OptiPlates (Nunc) for measuring fluorescence intensity of AF-488-labeled siRNA with VICTOR™ X5 multilabel plate reader (PerkinElmer, Waltham, MA, USA) attached with PerkinElmer 2030 manager software using λex = 490 nm and λem = 535 nm. Data were represented as mean ± S.D. of fluorescence intensity/500 mg of tissue mass after the values were blank-corrected using untreated group of mice for each organ.

### 2.16. Statistical Analysis

For MTT assay, a two-sample unpaired *t*-test was performed to analyze the difference between untreated and NP treated groups. Correlation coefficient was calculated for particle size with increasing calcium concentration. Two-sample paired t-test was performed for stability test. For in vivo biodistribution study, single factor ANOVA was run to compare the groups. Data were considered statistically significant at *p* value < 0.05.

## 3. Results

### 3.1. Pattern of Particle Growth for CA-in-DMEM and CA-in-DMB and HA

CA NPs were prepared with different concentrations of CaCl_2_·2H_2_O in either DMEM or in a mimicking buffer that contains only the key reactants—phosphate and bicarbonate. DMB also had D-glucose and NaCl to maintain smaller particle size [26]. HA particles were prepared with different concentration of Ca^2+^ in HEPES-buffered solution (HBS) as stated in Section 2.2. Absorbance at 320 nm was measured as an indication of NP formation. As shown in Figure 1A,B, both types of Cas (in DMEM and DMB) showed enhanced particles formation with increasing Ca^2+^ concentration, indicating Ca^2+^ is a driving force for the reaction. CA particles started to be formed with about 4-mM (For DMEM, 3.8 mM exactly with its 1.8-mM endogenous Ca^2+^) of total Ca^2+^ in both DMEM and DMB (Figure 1C). With the same concentration of total Ca^2+^, CA prepared in DMB had more particles which is reflected in its higher absorbance values. It indicates that the components of DMEM regulates particle growth in CA-in-DMEM. At higher concentration of Ca^2+^, the absorbance curve reached a plateau for both CA and HA (Figure 1B,D), which was due to the depletion of PO_4_^3−^ present in the medium/buffer at very low concentration (0.9 mM).

Microscopic images of CA-in-DMEM and CA-in-DMB (Figure 2) showed a similar trend of enhancement in particle growth. Immediately prepared CA-in-DMEM and CA-in-DMB were observed at 10× magnification under bright field using Olympus IX81 motorized inverted research microscope (Figure 2). With increasingly higher Ca^2+^ concentration, size and number of particles increased. Particles were visible under a microscope from 4-mM Ca^2+^ in DMEM and from 3-mM Ca^2+^ in DMB.

Formation of HA NPs increased up to 50 mM of Ca^2+^, maintaining a plateau up to 100 mM of Ca^2+^, followed by a drop in absorbance due to sedimentation (Figure 1D). For microscopic observation, HA particles were diluted in DMEM (Figure 3). Particles were small, aggregated and sedimented with increasing Ca^2+^ concentration in serum-free media. HA particles had a tendency to aggregate and sediment, which caused a drop in absorbance value after 100 mM of Ca^2+^. In serum-containing media, small HA particles were homogenously dispersed, which is probably due to binding of albumin and other serum proteins to the particles [27].

### 3.2. Measurement of Particle Size by Dynamic Light Scattering and FE-SEM

Size of CA increased proportionally with Ca^2+^ concentration. However, particle diameter increased only up to 350 nm with 7 mM of Ca^2+^ in DMEM (Figure 4A), while size of CA prepared in DMB drastically increased to 1500 nm with 4-mM Ca^2+^ and the size did not change significantly with further addition of Ca^2+^ (Figure 4B). HA NPs reached to a peak of 1200 nm with 15-mM Ca^2+^, after which size became smaller again (Figure 4C).

In FE-SEM images (Figure 5), all three NPs showed spherical shape, rough texture, with CA prepared in DMEM seeming to be the roughest. CA-in-DMEM and CA-in-DMB had a mixture of different sized particles—very small (<100 nm), medium (<300 nm) and big (400–600 nm). HA particles were uniform in size, ranging from 50 to 70 nm (Figure 5). By comparing the sizes measured in zeta sizer and FE-SEM—and from our visual observation—we assume that HA particles are small in size, with a high tendency of self-aggregation (shown in Figure 3).

### 3.3. Confirmation of Functional Groups by ATR-FTIR

Formation of CA-in-DMEM and CA-in-DMB was confirmed by the presence of transmittance peaks for CO_3_^2−^ (1641, 1480, 1415, 866 cm^−1^ and 1647, 1475, 1417 cm^−1^) and PO_4_^3−^ (1008, 562, 1014, 526 cm^−1^). ATR for HA particles gave peaks for PO_4_^3−^. For this nanoparticle, we also found few peaks for CO_3_^2−^ (1641, 874 cm^−1^), which could be due to incorporation of CO_3_^2−^ from atmospheric CO_2_. PO_4_^3−^ peak was wider for CA than that for HA due to CO_3_^2−^ incorporation in the lattice [28] (Figure 6).

### 3.4. pH Sensitivity of NPs

pH sensitivity of NPs was assessed through spectrophotometric analysis of absorbance in respective media or buffer of different pH values. For CA-in-DMEM and CA-in-DMB, 6 mM of total Ca^2+^ (endogenous + exogenous) was used to prepare particles, while Ca^2+^ concentration was 10-mM for HA. With decreasing pH, sharp decrease in absorbance values was observed for all three particles (Figure 7).

### 3.5. SiRNA-Binding Efficiency

Efficient NP-siRNA-binding is important to protect the siRNA from nuclease-mediated degradation as well as its unwanted dissociation from NPs before reaching the target. As shown in Figure 8A, over 85% of siRNA bound to the NPs CA-in-DMB and HA. Binding percentage was similar for both low and high concentration of siRNAs (1 nm, 2 nM and 10 nM). NPs-siRNA complex for these two particles were stable for lower concentration of siRNA even after a rigorous wash with pure water. However, binding efficiency reduced significantly for 10 mM of siRNA which was evident from Figure 8B. In contrast, siRNA-binding to CA-in-DMEM was concentration-dependent and significantly reduced with increasing siRNA concentration (Figure 8A). With a strong wash, the amount of bound siRNA came down to 30% (Figure 8B).

### 3.6. Stability Test for DMEM and CA Prepared in DMEM

Stability of NPs at various storage temperatures in clinical setting is important for their practical usage. We tested the stability of bicarbonated DMEM for instant particle formulation and pre-prepared CA-in-DMEM. DMEM was prepared and stored for 1, 7 and 30 days at +4 °C, −20 °C and −80 °C. In addition, nine sets of CA were prepared in DMEM and stored at three temperatures for the three above mentioned durations. CA NPs prepared with DMEM stored at three different temperatures—+4 °C, −20 °C and −80 °C, gave higher absorbance than that for freshly prepared NPs (Day 0). CA prepared with DMEM stored at +4 °C gave similar absorbance after three time points (Figure 9A). However, while compared to day-0 data, day-1 and day-7 data were significant at *p* < 0.05, data for day-30 were significant at *p* < 0.1 and seems to produce less particles at low Ca^2+^ concentration. The absorbance increased with time for DMEM stored at-20 °C and −80 °C (Figure 9B,C). While the chart shows a pattern of higher absorbance indicating more particles, data were statistically significant at *p* < 0.5, with an exception for Day 0 versus Day 30 data for DMEM stored at −80 °C, which was very significant at *p* < 0.01.

The pre-prepared CA stored at stored at +4 °C seemed to be degraded with time (Figure 9D), with significantly low absorbance for day-30 data. Pre-prepared CA stored at −20 °C and −80 °C gave significantly high absorbance from 1-mM Ca^2+^ compared to that for NP prepared with fresh buffer with statistical significance at *p* < 0.1 (Figure 9E,F). In a word, we can conclude that CA NP stored at +4 °C degrades with time. However, colder temperature supports increased particle formation.

### 3.7. Stability Test for DMB

We tested the stability of DMB at different temperatures for different durations. The buffer was taken out from storage and used to prepare NPs. Unlike DMEM, DMB was not stable when stored at +4 °C, −20 °C and −80 °C (Figure 10). The buffer started to lose its integrity by 24 h which was evident from the reduction in absorbance values of CA prepared using the stored buffer. Compared to day-0, absorbance was significantly low on Day 30 at *p* < 0.1 for DMB stored at −80 °C.

### 3.8. Stability Test for HBS

HBS buffer was stored at +4 °C, −20 °C and −80 °C for 1, 7 and 30 days and used to prepare NPs. HA prepared with HBS stored at 4 °C and −20 °C gave absorbance values, similar to that of freshly prepared particles (Figure 11A,B) and were not significantly different, except for day-30 data for HBS stored at +4 °C which was very significantly higher than day-0 data at *p* < 0.001. For HBS stored at −80 °C (Figure 11C), particle formation was significantly higher on Day 7 and Day 30, compared to that for fresh HBS on Day 0 (*p* < 0.01). We can consider that HBS can maintain its integrity at 4 °C for a duration smaller than 30 days and at −20 °C for 30 days.

### 3.9. Cytotoxicity of CA-in-DMEM, CA-in-DMB and HA

CA-in-DMB and HA did not contain any cell culture media to provide nutrients for cells to grow. Therefore, after formulation, these NPs were mixed with serum-containing DMEM. In this method of cellular treatment, we prepared all three NPs in 1 mL volume with total 6-mM Ca^2+^ for CA and 10-mM Ca^2+^ for HA. After fabrication, 100 μL of NPs was mixed with 900 μL of serum containing DMEM, before adding to MCF-7 cells seeded the day before. Both types of CA showed almost no cytotoxicity, whereas HA killed almost 34% cells (Figure 12).

### 3.10. Cellular Uptake of Fluorescence siRNA-Loaded CA and HA—Microscopic Analysis

CA-in-DMEM, CA-in-DMB and HA were formulated with 10 nM of fluorescent siRNA. After formulation, CA-in-DMB and HA were diluted five-fold because they were prepared in aqueous buffers without having any nutrients required for cells, whereas CA-in-DMEM was diluted too and considered as a control. After 4 h of treatment, fluorescence signal for the cells treated with all three NPs-bound siRNA was brighter than for the cells treated with free siRNAs (Figure 13). Alternatively, we prepared all three NPs in 100 μL and topped up with C-DMEM to 1 mL. In this experiment, with higher concentration of particles and siRNA, free siRNA and NP-siRNA in DMEM and DMB showed similar cellular uptake after 4 h (Appendix A). Uptake efficiency was lower for HA-siRNA compared to free siRNA in HBS (Appendix A). Substantial amounts of particles were visible in both serum-free and serum-containing buffer; however, HA uptake by the cells was not much. (Appendix A).

### 3.11. Identification of Spontaneously Formed PC on NPs, Via LC–MS

The common proteins found in the PC of three formulations of NPs include albumin, protease inhibitors from alpha-2 HS glycoprotein (AHSG) or fetuin-A gene and intermediate filament-keratin Type 1 cytoskeletal 10. CA-in-DMEM and CA-in-DMB also bound protease inhibitor SPP24 and hemoglobin. Only CA-in-DMB bound protease inhibitors from serpin superfamily, transferrin, apolipoproteins, clotting factors and immunoglobulins. Interestingly, between CA-in-DMEM and CA-in-DMB, the latter bound more proteins of various types. Besides the common proteins shared by all three NPs, HA bound immunoglobulin-binding protein and some different keratins. All the proteins are listed in Table 1. Most of the proteins detected were acidic with pI value below 7.5 (Figure 14).

### 3.12. Biodistribution of NPs in a Mouse Model of Breast Cancer

For a biodistribution study, CA-in-DMEM and CA-in-DMB were prepared with total (exogenous + endogenous) 6 mM of Ca^2+^ in 100 μL volume. HA was prepared with 10-mM Ca^2+^ in 100 μL HBS. A 0.5-μM fluorescence siRNA was complexed with each type of NPs. After 4 h of intravenous delivery of siRNA-loaded NPs or free siRNA, mice were sacrificed. The organs were collected, and fluorescence was measured. The additional fluorescence from the supernatant of lysed organs of treated animals in comparison with that from the organs of untreated animals was caused by the siRNAs taken up by the organs (Figure 15).

Seven organs were studied for the accumulation of free or NPs-bound siRNAs. Free siRNA was detected in all organs except tumor. Among the three different NPs, CA (both in DMEM and DMB) showed significant uptake by tumor, although fluorescence intensity was almost ten-fold for CA-in-DMB. CA-in-DMEM did not deposit in the heart, but CA-in-DMB did. Neither of the NPs reached the brain.

The signal was negative from kidney for all three NPs. On the contrary, systemic delivery of NPs loaded with siRNA resulted in substantial accumulation in the liver and spleen and organs of the reticuloendothelial system. Apart from free siRNA, CA-in-DMB and HA reached the lungs. Among the NPs and free siRNA there was no statistically significant difference in their levels in spleen, heart, lung and liver.

## 4. Discussion

### 4.1. Pattern of Particle Growth, Size and SiRNA-Binding

Turbidity or absorbance data and microscopic images (Figure 1, Figure 2 and Figure 3) show that particle growth enhanced with increasing Ca^2+^ concentration which is a driving force for the reaction (of particle formation) to happen. For further in vitro and in vivo studies, Ca^2+^ concentrations of 6 mM and 10 mM were chosen for CA and HA, respectively due to the similar amounts of particles apparently observed for both NPs under the optical microscope. ATR-FTIR data (Figure 6) identified the functional groups of CA and HA in the form of the peaks for CO_3_^2−^ and PO_4_^3−^. HA spectrum showed peaks for CO_3_^2−^ along with PO_4_^3−^. Though the HBS buffer did not contain any carbonate salt, this functional group was likely to be incorporated from environmental CO_2_.

SEM image for dried, serum-free sample of CA prepared in DMEM and DMB with the same amount of Ca^2+^ showed formation of similar sized particles ranging from 100–600 nm with similar round shape. However, the same NPs in 10% serum-containing media showed very different sizes measured with ZetaSizer: 74 nm for CA-in-DMEM and 1686 for CA-in-DMB. Similar pattern was observed for HA. With 10-mM Ca^2+^, HA size was 50–70 nm observed in SEM, while in 10% serum-containing media it was measured 737 nm with ZetaSizer (Figure 4 and Figure 5). Therefore, the overall observation was that CA prepared in DMEM reduced size in contact with serum, whereas the other two particles, CA-in-DMB and HA, prepared in buffers, got bigger in serum-containing media. Increase of the size of HA in serum-containing media was reported by other researchers too [29]. The distinctive PC formed around each type of NPs may play a crucial role to determine their sizes. While we observed particle size exceeding 1 µm for CA-in-DMB and HA, CA-in-DMEM was <500 nm even when Ca^2+^ concentration was increased (Figure 4). We assume that the amino acid and vitamins present in DMEM have regulatory effects on particle growth. Effect of DMEM was also evident on CA’s ability to bind siRNA. While CA prepared in DMB and HA stably bound siRNAs, CA-in-DMEM lost bound cargo upon wash (Figure 8). One potential reason of dissociation of siRNA from CA-in-DMEM could be the shifting of equilibrium for particle synthesis to the reverse direction. Particles prepared in DMEM probably had impurities such as vitamins, amino acids, etc., which were removed in contact with excess pure water, resulting is particle dissolution.

All three types of particles were pH-sensitive and dissolved completely at pH 6.5 (Figure 7). We can infer that these particles can be dissolved quickly to release the bound cargo of siRNAs or other therapeutics in the acidic microenvironment of tumor tissues, which can be as low as pH 6.44 [30].

### 4.2. Stability of the Buffers and CA-in-DMEM

We investigated the stability of prepared NPs of CA-in-DMEM and the media/buffers of DMEM, DMB and HBS over 1, 7 and 30 days at 4 °C, −20 °C and −80 °C. We found that CA prepared with stored DMEM produced more particles. Particle number grew further in pre-prepared CA suspension too, except for CA stored at four degrees Celsius (Figure 9). We assume that the endogenous salts including inorganic phosphate (0.9 mM), bicarbonate (44 mM) and Ca^2+^ (1.8 mM) in DMEM were more readily available (probably due to less association with existing organic molecules in media) to react with exogenous Ca^2+^, resulting in more particle formation. DMB buffer was not stable at all (Figure 10); in fact, this buffer changed pH even at room temperature in a few hours (data not shown), which could be due to loss of HCO_3_^−^ in the form of CO_2_. HBS buffer was found stable (Figure 11), since it contained neither DMEM ingredients (e.g., vitamins, amino acids) and bicarbonate.

### 4.3. Potential Roles of PC Components

Upon systemic delivery, NPs come into contact with serum proteins which immediately wrap the particles forming a PC. Composition of PC depends on chemical composition, size, charge and other physicochemical properties of NPs and also on the abundance of the proteins in blood [31]. PC in vivo can be remodeled with the change of proteome in many disease conditions [32]. Typically albumin, immunoglobulins, apolipoproteins and clotting factors are common constituents of NPs’ PC [33], which was the case for our NPs as well (Table 1). PC composition of all three different NPs shared albumin and protease inhibitors of fetuin family. Interestingly, between CA-in-DMEM and CA-in-DMB, the latter bound more proteins of various types—and unlike CA–HA bound a number of keratins and immunoglobulin-binding proteins. Along with different chemical composition, this conspicuous difference in PC composition among the three NPs could be attributed to different types of media/buffer they were formulated in, which differ in the levels of several salts, amino acids and vitamins [34] (Table 1). This idea was corroborated by a study which reported difference in interactions between proteins and NPs in DMEM and RPMI for the same gold NPs and thus their different impact on cells [35].

Albumin, the most abundant and an acidic protein in blood, has affinity for HA and CA [36]. As a dysopsonin, albumin reduces hepatic clearance of NPs and increases their circulation time [37]. Albumin can be absorbed by alveolar epithelial cells [38]. Having similar composition as bone mineral hydroxyapatite, all three types of particles also shared similar PC components which resemble the proteins in calciprotein particles (CPP) and the bone mineral–protein complex formed in our bodies to prevent unwanted precipitation of salts in blood vessels or in other undesired locations during the process of bone matrix metabolism [27]. Fetuin, an alpha-2 HS glycoprotein (AHSG) and other members from the AHSG family work as mineral chaperones to stabilize, transport and excrete bone minerals (calcium and phosphate) from the body. While most of the detected proteins were acidic, SPP 24, an endopeptidase inhibitor in the PC of CA, was basic. This protein is reported to be involved in bone mineral-fetuin complexes [39]. Fetuin-A protease inhibitors not only work as a carrier of Ca^2+^ and PO_4_^3−^, but also facilitate endocytosis of NPs by macrophages [40]. Fetuin bound particles are taken up and cleared by reticuloendothelial system. In this procedure of clearance, liver and spleen play the major role with the scavenging activity of their Kupffer cells and marginal zone macrophages, respectively [41].

Both CA-in-DMEM and CA-in-DMB bound to hemoglobin alpha and beta subunits, globin domain-containing protein and hemopexin. CA-in-DMB also bound two other heme/iron-binding proteins, transferrin and alpha 1-microglobulin. A study on multi-wall carbon nanotubes (MWCNT) coated with hemoglobin and transferrin observed enhanced uptake of MWCNT in rat mesothelial cells through transferrin receptor [42]. Interestingly, transferrin receptors were found to be overexpressed in breast carcinoma as well as in ovary and lung [43]. We predict that presence of hemoglobin and transferrin in the PC of CA played a role for its ability to deliver fluorescence siRNA to the breast tumors in our animal model (Figure 15).

Serine protease inhibitors from serpin superfamily, alpha 1-antitrypsins and alpha 1-antichymotrypsins, were identified in the PC of CA-in-DMB [44,45]. This class of acute phase proteins were also noticed to interact with other inorganic NPs such as silver NP, BaSO_4_ NP, ZnO NP [46,47]. Coating of NPs with these acute phase proteins facilitates their uptake by the hepatocytes via serpin-enzyme complex receptor [48]. Biodistribution of CA-in-DMB in lung could be due to the presence of serpins in its PC, as substrates for these protease inhibitors, neutrophil elastase for antitrypsin and cathepsin G for antichymotrypsin, function in lung airway secretion [49].

CA-in-DMB also bound apolipoprotein A2, prothrombin and immunoglobulins. PC analysis of iron oxide NPs revealed that these high affinity and slow exchanging proteins bind strongly on particle surface and can work as linkers to bind albumin, an abundant yet low affinity dysopsonin protein [34,37,50]. At the same time apolipoproteins and immunoglobulins opsonize and result in phagocytosis of the NPs larger than serum proteins, which are 20–30 nm, a phenomenon observed in case of gold NPs [51].

PC of HA NP includes immunoglobulin-binding proteins and several keratins. While HA bound quite a number of keratins, PC of CA-in-DMEM and CA-in-DMB bound only keratin Type I cytoskeletal 10. Presence of keratin in the PC was reported for some other NPs such as multi-wall carbon nanotubes (MWCNT) [42] and silver NPs [46]. Keratins provide stability and integrity as a part of cytoskeleton in epithelial cells. They also contribute in stress management, apoptosis and wound healing [52]. While few researchers described the presence of keratin in blood [53,54], little was revealed about the normal level of different keratins in blood, specially that in mouse and how they may impact the uptake of NPs by the organs.

### 4.4. NPs-Mediated SiRNA Delivery In Vitro and In Vivo

CA prepared in both DMEM and DMB showed similar siRNA delivery efficiency in MCF7 breast cancer cells, however we noticed poor cellular uptake of HA bound siRNA (Figure 13). Although particle number was seemingly sufficient for HA (Appendix A), we did not notice cellular uptake of fluorescent siRNA even after increasing siRNA concentration to 10 nM (Appendix A). HA showed high cytotoxicity (34%) upon a 48 h treatment (Figure 12). This type of NPs could not enter the cells, instead it deposited on the cell surface and thus likely caused cell death.

Similar to in vitro data, uptake of HA-bound fluorescent negative siRNA was poor in the tumor tissue; instead, we observed deposition in heart, along with lung, liver and spleen. Between the two CAs, only CA-in-DMB was detected in the heart and lungs. None was detected in kidney and brain. However, among the four treatments tested (free siRNA and siRNA bound to particles of CA-in-DMEM, CA-in-DMB and HA), only the CAs accumulated in the tumor (Figure 15).

The size of NPs could be a major determinant of their fate. An early study on hydroxyapatite by Martin and Brown showed that HA formation was inhibited by serum; however, the inhibitory effect of serum was weaker on carbonated HA [55]. We observed directly proportional size reduction with serum concentration (data yet to be published) which is probably due to the change in dynamic PC with increasing serum concentration. In blood, where serum concentration is about 50%, the particle size is supposed to reduce. We noticed that our CA formulation yields a mixture of particles of different sizes. Since the small sized particles could be excreted via renal system very rapidly, they were not detected in kidneys after four hours of intravenous administration. However, from the biodistribution data, the reticuloendothelial system seemed to be the major mechanism for the clearance of our particles. Deposition in the spleen and liver could be attributed to the fetuin-A and serpin protease inhibitors in the PC of these particles [41,48].

Coating of albumin on the NPs may play a role for lung deposition; however, it cannot be said conclusively as CA-in-DMEM was not accumulated in the lungs. Despite having very different PC composition, HA and CA-in-DMB were similarly distributed in heart and lungs, while the former did not accumulate in tumor. Further investigation may reveal the underlying reason for this observation, especially if keratin in the PC plays any role for the poor uptake of HA by tumor. Free siRNA was detected in all the organs except the tumor. The inability of free siRNA to enter tumor tissue may be attributed to the acidic microenvironment, leading to its degradation [56].

## 5. Conclusions

CA has already been established as a successful siRNA delivery system in preclinical trials in animal model. Here we formulated CA in a DMEM-free media and also explored the potential of HA prepared in a similar method. Our findings suggest that CA NPs prepared in a DMEM-free media bind siRNA more strongly and deliver the cargo to breast tumor tissue 10 times more efficiently. CA-in-DMB deposited in heart, which could be a potential problem if the delivered siRNA targets a gene vital for heart functions. HA was stable and able to load high amount of siRNA, but siRNA delivery efficiency was poor both in vitro and in vivo. We infer that PC composition rendered CA-in-DMB superior delivery efficacy over CA-in-DMEM. In addition, it could be the reason for HA’s poor performance. Future studies should be carried out to elucidate the roles of specific protein components of PC in directing the NPs to the tumor and blocking their off-target distribution to other organs.

## Figures and Tables

**Figure 1 jfb-11-00063-f001:**
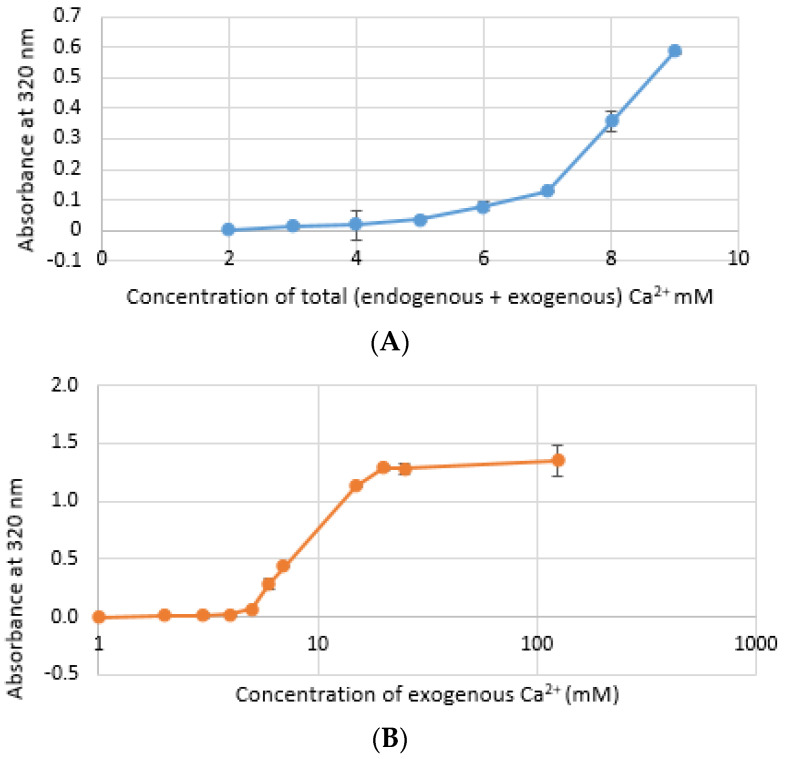
Turbidity of NPs at 320 nm as an indication of particle formation. (**A**) Carbonate apatite (CA) prepared in Dulbecco’s modified Eagle’s medium (DMEM). Particles formation increases with increasing Ca^2+^ concentration; (**B**) similar increasing trend was noticed for CA prepared in DMEM-mimicking buffer (DMB); (**C**) comparison of turbidity between CA prepared in DMEM and DMB; (**D**) HA started to form from 5 mM of Ca^2+^ in HBS and particle growth continued to increase up to 100 mM of Ca^2+^, followed by a drop.

**Figure 2 jfb-11-00063-f002:**
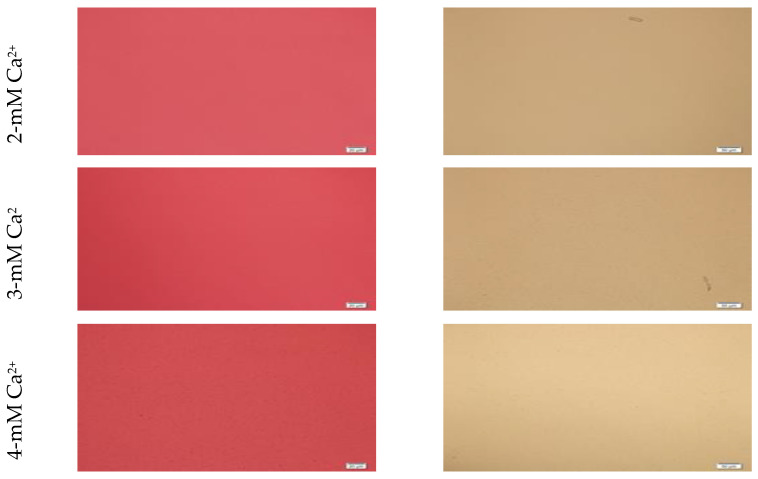
Microscopic images of CA-in-DMEM and CA-in-DMB. Particles prepared with different concentrations of total Ca^2+^. Visible particles started to form with a minimum 3 mM of Ca^2+^ in DMB, while in DMEM, they started at 4 mM of total Ca^2+^.

**Figure 3 jfb-11-00063-f003:**
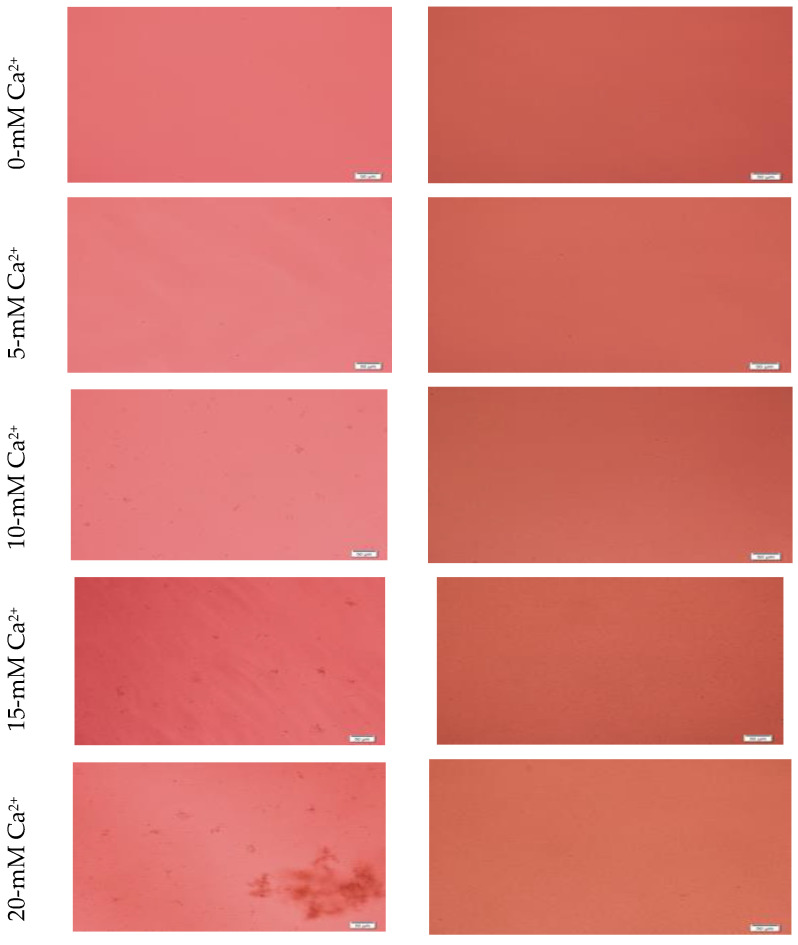
Microscopic images of hydroxyapatite (HA). After fabrication, nanoparticles (NPs) were observed under a microscope with or without adding serum. While particles were aggregated in the absence of serum (left column), they were homogenously dispersed upon addition of serum (right column).

**Figure 4 jfb-11-00063-f004:**
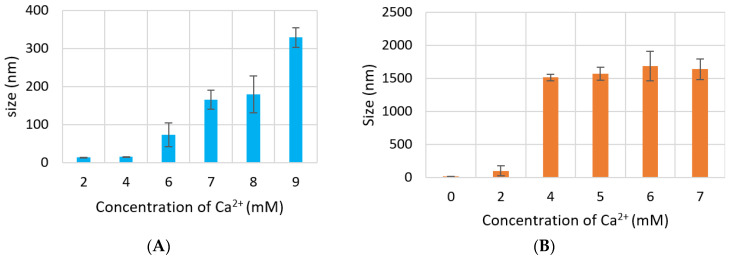
Size of NPs with increasing Ca^2+^ concentration measured by zeta sizer. (**A**) CA-in-DMEM; (**B**) CA-in-DMB; (**C**) HA. Correlation between calcium concentration and particle size was calculated for each NP. Correlation coefficient values for CA-in-DMEM and CA-in-DMB were 0.90 and 0.92, respectively which indicate strong positive correlation. A 10% positive correlation was calculated for HA (correlation coefficient 0.1).

**Figure 5 jfb-11-00063-f005:**
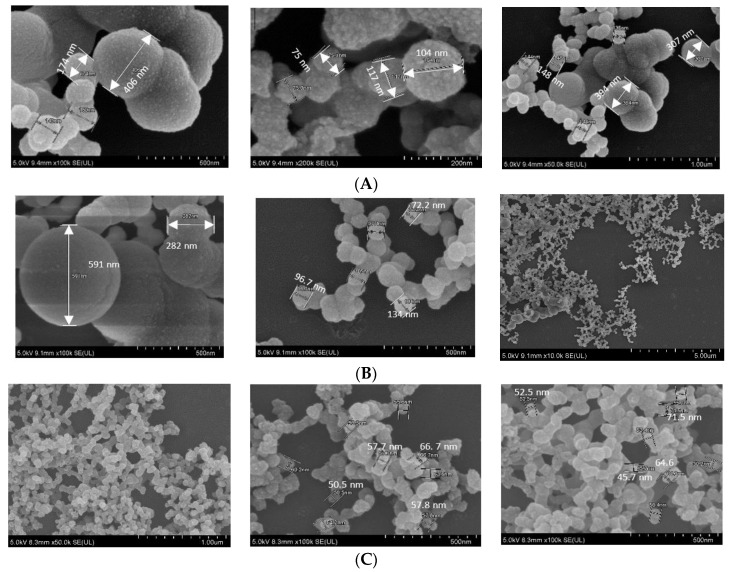
SEM image of NPs. (**A**) CA-in-DMEM; (**B**) CA-in-DMB; (**C**) HA.

**Figure 6 jfb-11-00063-f006:**
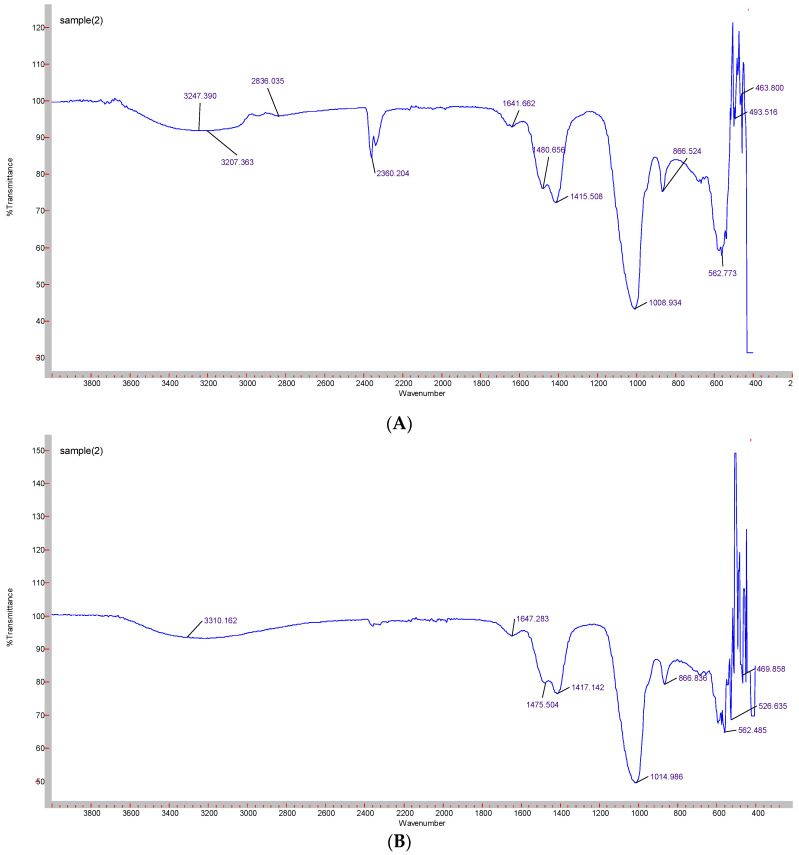
ATR-FTIR spectra to determine the functional groups present in the NPs. (**A**) CA-in-DMEM; (**B**) CA-in-DMB; (**C**) HA.

**Figure 7 jfb-11-00063-f007:**
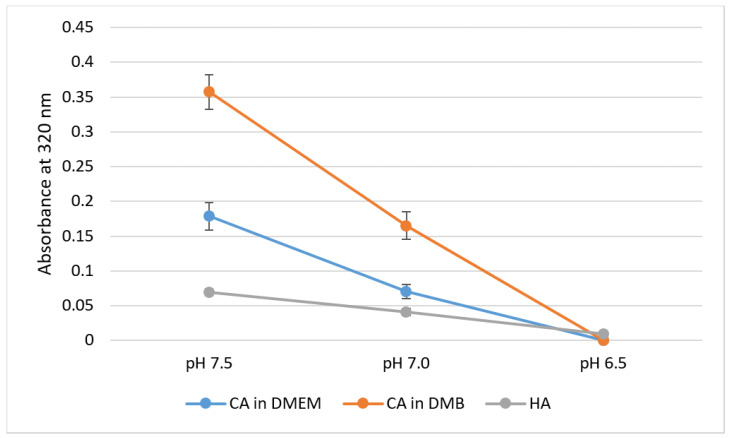
pH-dependent dissolution of CA-in-DMEM and CA-in-DMB and HA. Particles prepared in DMEM/buffers of pH 7.5 in 200-μL volumes and afterward mixed with 800 μL of DMEM/buffers of different pH values. Absorbance measured at 320-nm wavelength. Data show increased particles dissolution with decreasing pH.

**Figure 8 jfb-11-00063-f008:**
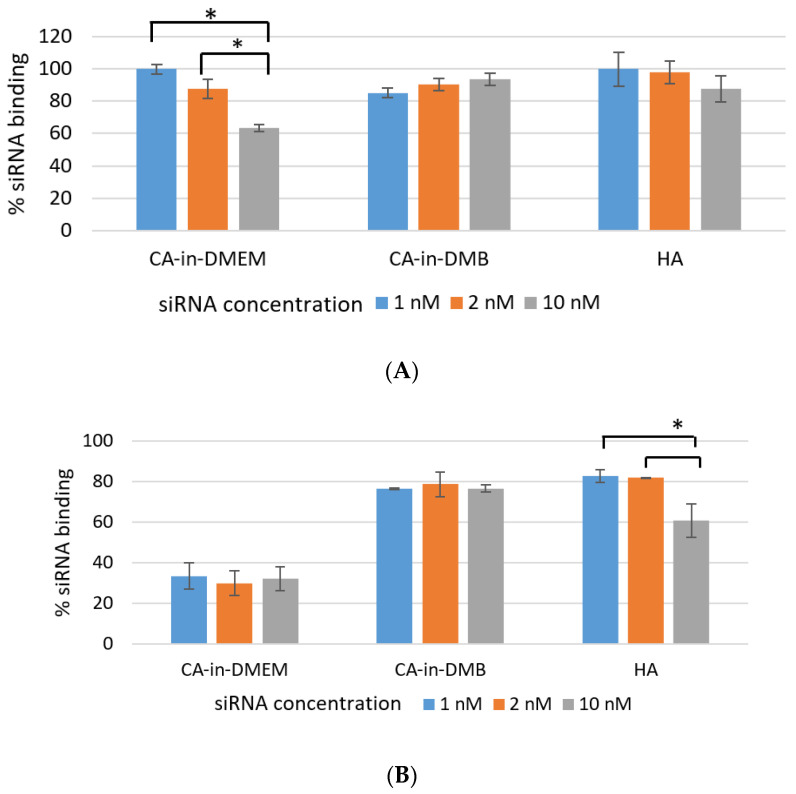
Efficiency of siRNA-binding to CA-in-DMEM, CA-in-DMB, and HA. After formulation, particles complexed with siRNA were centrifuged, followed by removal of supernatant. The pellets were (**A**) directly dissolved in 10-mM EDTA or (**B**) washed with water by another 15-min centrifugation before dissolving in EDTA. Concentrations of the siRNA bound to the particles were calculated from the formula in Section 2.9. CA-in-DMB and HA showed stable and high binding to siRNA. * indicates significant difference at *p* < 0.05.

**Figure 9 jfb-11-00063-f009:**
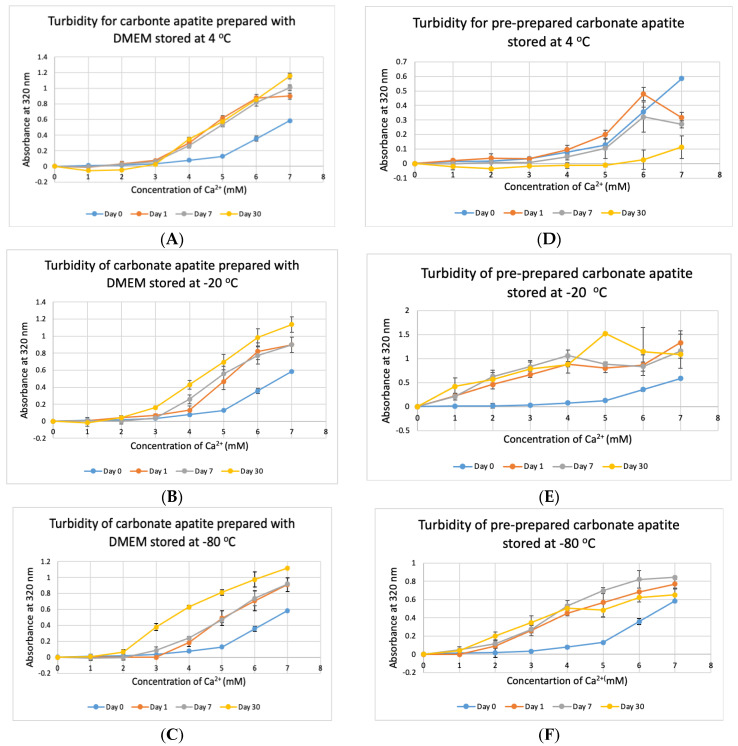
Stability of DMEM and CA-in-DMEM stored at different temperatures for different durations. (**A**–**C**) Absorbance of CA prepared with DMEM that was stored at 4 °C, −20 °C and −80 °C for 1, 7 or 30 days; (**D**–**F**) absorbance measured for prepared CA stored at 4 °C, −20 °C and −80 °C for 1, 7 or 30 days. Two-sample paired T-test performed for statistical analysis. For (**A**–**C**), compared to Day 0, all data were significant. For preprepared CA, compared to Day 0 data, particle formation was significantly higher when stored at −20 °C and −80 °C (**E**,**F**).

**Figure 10 jfb-11-00063-f010:**
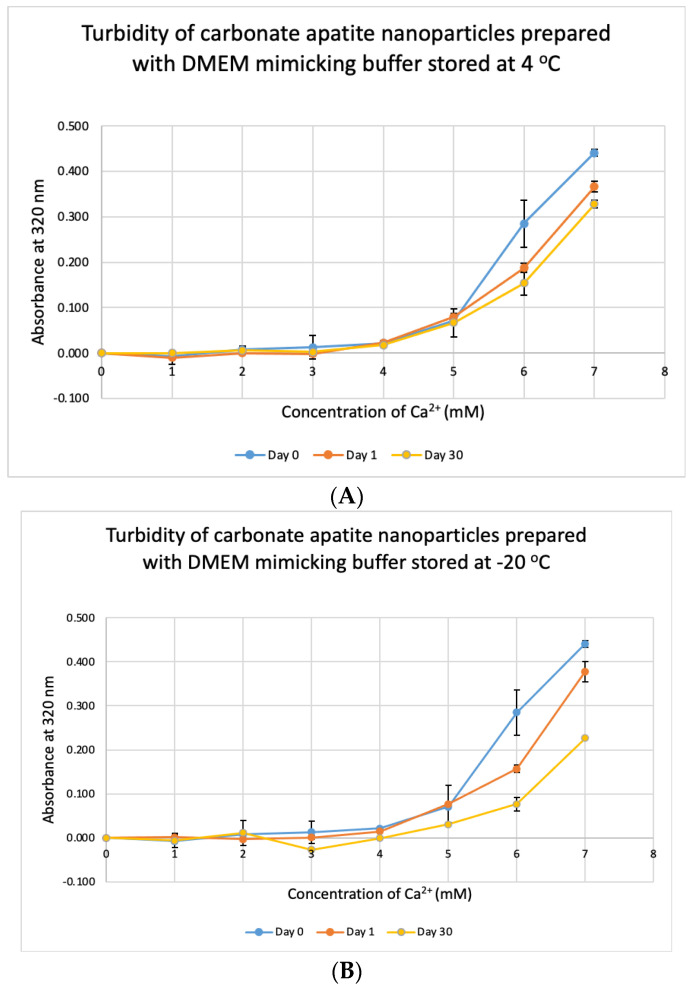
Stability of DMB stored at different temperatures for different durations. (**A**–**C**) Absorbance of CA prepared in DMB stored at 4 °C, −20 °C and −80 °C for 1 or 30 days, respectively. Two-sample paired *t*-test performed for statistical analysis.

**Figure 11 jfb-11-00063-f011:**
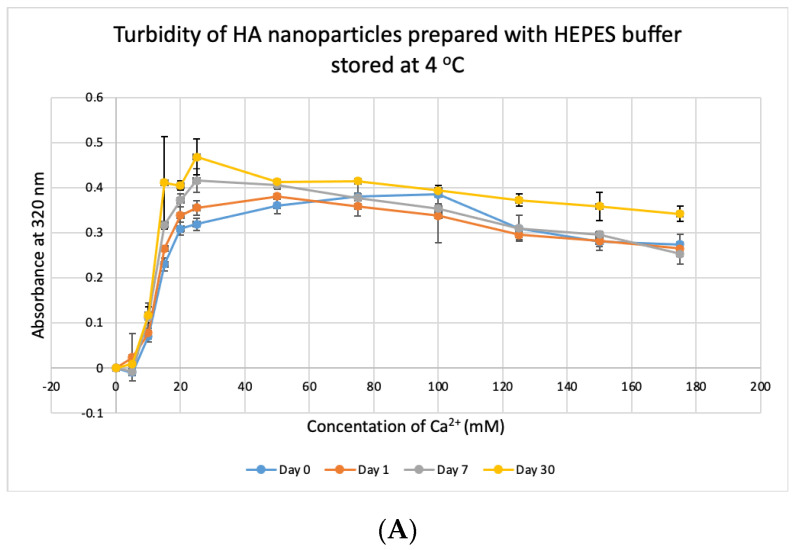
Stability of HBS stored at different temperatures for different durations. (**A**–**C**) Absorbance of HA prepared with HBS that was stored at 4 °C, −20 °C and −80 °C for 1, 7 or 30 days, respectively. Two-sample paired *t*-test performed for statistical analysis. Compared to freshly prepared HA, significantly more particles were formed with HBS stored at 4 °C for 30 days and at −80 °C for 7 and 30 days.

**Figure 12 jfb-11-00063-f012:**
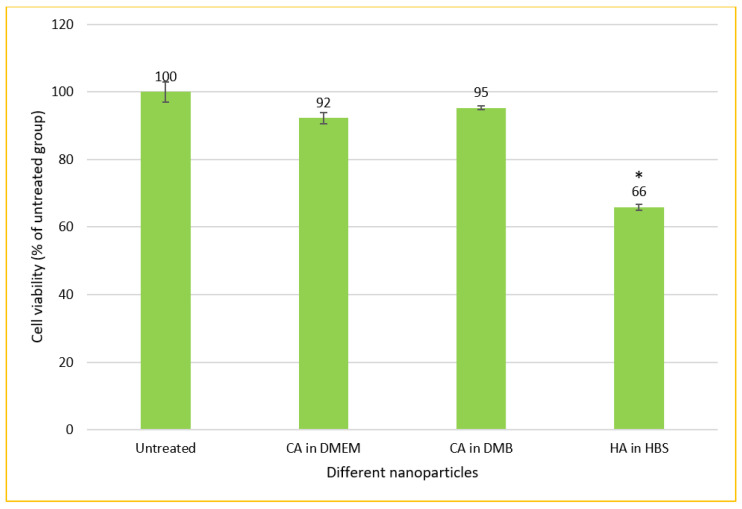
In vitro cytotoxicity of CA and HA in MCF-7 cell line. NPs prepared in 1-mL volume and diluted 10 times before adding to the cells seeded the day before. MTT assay performed after 2 days of treatment. * Compared to untreated groups, HA decreased cell viability significantly at *p* < 0.05 level.

**Figure 13 jfb-11-00063-f013:**
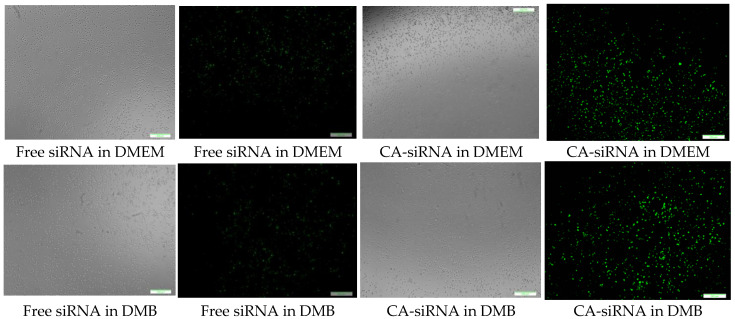
Cellular uptake of NPs loaded with Alexa Fluor 488 neg siRNA. All NPs prepared in 1 mL with 10 nM of siRNA and diluted 5-fold in C-DMEM. After 4 h, cells were washed with 5-mM EDTA and PBS before viewing under microscope. Row 1: CA-in-DMEM; Row 2: CA-in-DMB; Row 3: HA. Scale 50 μm.

**Figure 14 jfb-11-00063-f014:**
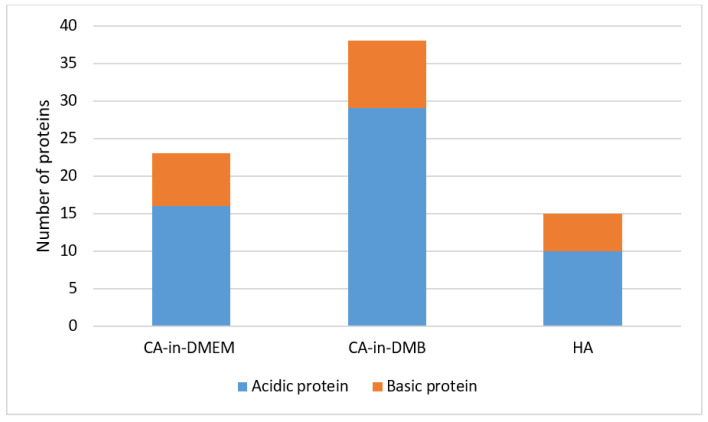
Classification of proteins regarding pH value in the protein corona of NPs. Proteins with pI value below 7.5 are acidic and those with pI value above 7.5 are basic in nature.

**Figure 15 jfb-11-00063-f015:**
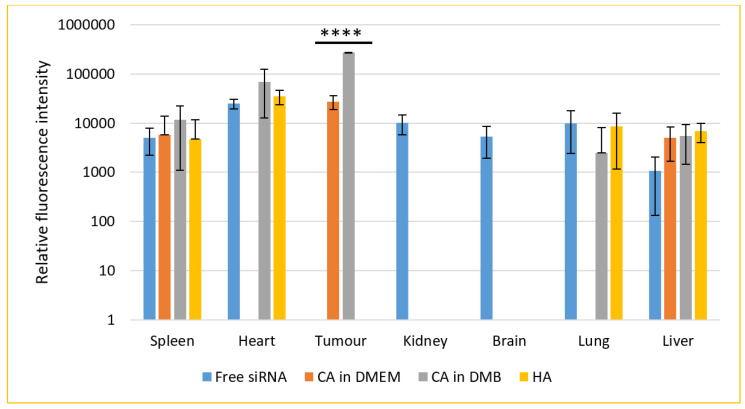
Biodistribution of CA-in-DMEM, CA-in-DMB and HA in a mouse model of breast cancer. NPs loaded with fluorescence siRNA were injected into tumor-bearing mice via tail vein. After 4 h, the animals were sacrificed and organs were collected and lysed. The tissue lysates were centrifuged at 8000 rpm for 20 min. Fluorescence was measured for the supernatants. Each group contained 5 animals. Background fluorescence was measured using the organ and tumor lysates of untreated mice and the measured values were deducted from those of treated groups. Data were presented as average ± S.D. and the graph was plotted in logarithmic scale. **** Values for two CAs accumulated in tumor tissues were significantly different at *p* < 0.0001 level. No NPs were detected in kidneys or brain. CA-in-DMEM was not detected in heart and lungs. Spleen and liver showed presence of free siRNA as well as NPs. However, there was no significant difference.

**Table 1 jfb-11-00063-t001:** List of identified proteins from the protein corona (PC) of CA-in-DMEM, CA-in-DMB and HA.

Accession	Mass (Da)	pI	Identified Protein	Molecular Function	Biological Process	CA-in-DMEM	CA-in-DMB	HA
Q546G4	68693	5.75	Albumin 1	Albumin gene family	Carrier protein	✓	✓	✓
A0A0G2JGM6	14173	5.46	Vitamin D-binding protein			·	✓	·
Q3TIU3	37298	5.94	Alpha-2-HS-glycoprotein	Protease inhibitor	carrier protein, bone matrix metabolism, binds Ca and phosphate	✓	✓	✓
Q3UEK5	37280	6.04	Alpha-2-HS-glycoprotein			✓	✓	✓
Q3UEK9	37326	6.04	Alpha-2-HS-glycoprotein isoform CRA_a			✓	✓	✓
Q711L0	22677	8.9	Secreted phosphoprotein 24	Protease inhibitor	Bone remodelling	✓	✓	∘
D3YTY9	53205	4.95	Kininogen-1	Protease inhibitor	Blood coagulation	·	✓	∘
A0A0R4J0I1	46673	5.04	MCG1051009/Serine protease inhibitor A3K	Alpha 1-antichymotrypsin	Acute phase	∘	✓	∘
A0A0R4J0X5	45854	5.32	Alpha-1-antitrypsin 1-3	Alpha 1-antitrypsin		∘	✓	∘
A0A0A0MQA3	48796	5.98	Alpha-1-antitrypsin 1-1			∘	✓	∘
Q3KQQ4	45896	5.31	Serpina1a protein			∘	✓	∘
A0A1B0GS57	21329	6.58	Hemopexin	Heme/iron binding	Heme/iron binding and oxygen transport	✓	✓	·
Q8K1U6	31047	7.64	Hemopexin			✓	✓	·
D3YYR8	25676	8.34	Transferrin			·	✓	·
A0A0C6E3V3	51022	6.82	HC protein/alpha-1-microglobulin			·	✓	·
Q9QWJ3	12907	6.78	Alpha-1-globin	Hemoglobin alpha subunit		✓	✓	·
Q91VB8	15112	7.97	Alpha globin 1			✓	✓	·
A8DUV3	15085	7.97	Alpha-globin			✓	✓	·
D0U270	15840	7.13	Beta-globin	Hemoglobin beta subunit		✓	✓	·
A8DUP5	15894	7.09	Beta-globin			✓	✓	·
A8DUK0	15840	7.13	Beta-globin			✓	✓	·
A8DUP7	15826	7.13	Beta-globin			✓	✓	·
Q61650	5690	4.65	Beta-globin			✓	✓	·
A8DUM2	15780	7.13	Beta-globin			✓	✓	·
D4N6U4	15880	7.13	Beta-globin			✓	·	·
B1Q450	15840	7.13	Beta-globin			✓	·	·
D0U269	15854	7.13	Beta-globin			✓	·	·
Q9CY06	15112	7.97	Globin domain-containing protein	Heme, iron, binding		✓	✓	·
Q9CY10	15202	8.95	Globin domain-containing protein			✓	✓	·
Q8BPF4	15181	8.72	Globin domain-containing protein			✓	✓	·
Q6LD55	11319	6.57	APOAII	Lipid binding and transport	Opsonin	·	✓	·
A7YL62	11291	6.57	Apolipoprotein A-II			·	✓	·
H7BX99	70212	6.04	Prothrombin	Clotting factor	Blood coagulation	·	✓	·
Q3TJ94	70269	6.04	Prothrombin			·	✓	·
A0A075B5P6	50063	6.6	Ig mu chain C region	Immunoglobulin	Complement activator, Opsonin	·	✓	·
A0A075B6A0	52618	5.99	Ig mu chain C region			·	✓	·
A0A075B5P4	35752	7.2	Ig gamma-1 chain C region			·	✓	·
A0A0A6YWR2	43434	6.02	Ig gamma-1 chain C region			·	✓	·
Q99LC4	51008	6.6	Igh protein			·	✓	·
I6L985	51976	8.12	Igh protein			·	✓	·
E9Q9C6	275239	4.86	Fc fragment of IgG-binding protein	Immunoglobulin binding protein		·	·	✓
E9Q0B5	275224	4.87	Fc fragment of IgG-binding protein			·	·	✓
A2A513	57041	5	Keratin type I cytoskeletal 10	Epithelial intermediate filament	mechanical support	✓	✓	✓
Q9CV72	30011	5.4	IF rod domain-containing protein			·	·	✓
Q3UV11	59526	8.32	Keratin type II cytoskeletal 6B			·	·	✓
Q0VDR7	60273	8.33	Krt6b protein			·	·	✓
Q32P04	61767	7.59	Keratin 5			·	·	✓
Q08EK4	61302	7.73	Keratin 77			·	·	✓
B1AQ78	44542	5.28	Keratin 19			·	·	✓
B1AQ77	49494	4.79	Keratin 15 isoform CRA_a			·	·	✓
B2RTP7	70923	8.26	Krt2 protein			·	·	✓

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
