# Peer review of "Carbonate Apatite and Hydroxyapatite Formulated with Minimal Ingredients to Deliver SiRNA into Breast Cancer Cells In Vitro and In Vivo"

_jfb, 2020, doi:10.3390/jfb11030063_

Round 1

Reviewer 1 Report

Dear Sirs

I have several major comments on your manuscript:

  1. You used only one breast cancer cells. I understand the logistic problem but to publish your story you need to show the same effect using at least 2 cell lines of breast cancer. Also, you need to test the effect of your designed nano in3D BrCa cells
  2. 3. There no negative control using in the study such as fibroblast, breast epithelial cells etc. It is important to show it.
  3.  There are no tests conducted using serum if I understand correctly (DMEM sample means DMEM sample only?). You binding affinity may be an artificial.

Author Response

Reviewer 1 comments:

  1. You used only one breast cancer cells. I understand the logistic problem but to publish your story you need to show the same effect using at least 2 cell lines of breast cancer. Also, you need to test the effect of your designed nano in3D BrCa cells.
  2. There no negative control using in the study such as fibroblast, breast epithelial cells etc. It is important to show it.

Answer: In our lab, carbonate apatite is considered as a control to understand the efficiency of new particles, and sometimes other cell lines are used too. There are two goals behind this study. Firstly, to formulate a DMEM free carbonate apatite, which we refer as carbonate apatite in DMEM mimicking buffer or CA-in-DMB, and secondly, to answer if hydroxyapatite prepared in a similar manner be able to deliver siRNA. In this study, we found that CA-in-DMB performs better than CA-in-DMEM and a formulation method similar to that for CA-in-DMEM/CA-in DMB is unable to prevent the aggregation of HA and thereby needs further research to improve. As our ultimate goal is the clinical usage, we performed biodistribution assay. We consider that in vivo performance is more important and in vitro efficiency of an NP does not necessarily predict its success in vivo.

We agree with the reviewer that using a non-cancer cell line and multiple cancer cell lines unquestionably establishes the credibility of any NP and we surely will try our best to do it for our future research. However, addressing this issue for this paper will take months to complete and even longer due to the current social distancing rules in lab. It is also noteworthy that the final goal would be tumor-specific delivery of the siRNA-loaded nanoparticles either via passive targeting harnessing enhanced permeability and retention (EPR) effect of tumor microenvironment, or active targeting through coating a tumor receptor-specific ligand on nanoparticle surface. This is how off-target distribution of the nanoparticles  could be significantly minimized.

  1. There are no tests conducted using serum if I understand correctly (DMEM sample means DMEM sample only?). You binding affinity may be an artificial.

Answer: The reason for not using FBS for binding assay is that FBS gives fluorescence at 488 nm which interferes with Alexa flour 488 siRNA. To rule out fluorescence from unbound or loosely bound siRNA with NP, the samples were vigorously washed by centrifugation which reduced % bound siRNA to 30% for CA-in-DMEM. However, even after wash, % siRNA binding did not change for CA-in-DMB and HA (Figure 8).

Reviewer 2 Report

The manuscript presents the synthesis and characterization of carbonate apatite and hydroxyapatite nanoparticles for delivery of siRNA. The subject of the manuscript is suitable for this journal, the methods and results are well written. I recommend for publication after corrections of minor errors in text editing.

Author Response

Thank you very much.

Reviewer 3 Report

In this manuscript, the author developed simplified methods of formulating Carbonate apatite (CA) in DMEM and a DMEM-mimicking buffer and hydroxyapatite (HA) in a HEPES-buffered solution. It is beneficial to the development of new drug delivery systems. This study also demonstrated that CA is more efficient than HA for siRNA delivery to the tumor and it is more efficient in a buffer containing only the mere constituents. This manuscript may be potentially interesting, however, there are a number of issues that should be addressed and clarified in order to increase the significance of the work.

Major points:

  1. In the part of “Cytotoxicity of CA-in-DMEM, CA-in-DMB and HA”,controls of other breast cancer cell lines and several normal cells are needed. It's better to be able to verify the toxicity effect on mice.
  2. In this paper, authors employed subcutaneous tumor model. However, to be more convincing, orthotopic breast cancer model should also be implemented. And should clearly clarify the data directly support the result “tumors were approximately 100 mm3 in volume”.
  3. Why only used 4 hours treatment? Better set a series of time points, to observe the accurate dynamic biodistribution of NPs in the organs.
  4. Better at least six mice were used to animal test.

Minor points:

  1. “The of size of ” should be corrected to “The size of ” (Page 1, line 39)
  2. In the material and method part, the information of where 4T1 cells are purchased from, and how to culture, is missing.
  3. Description of statistics analysis is not clear enough.
  4. The rulers in Figure 2 , 3 and 13 is not clear, and one ruler in Figure 13 is not placed in the bottom right-hand corner(Page 10-11, line 326-332).
  5. Lack of statistics analysis in Figure 4, 8-11 (Page 12,line 338-339), (Page 15-16, line 373-378), (Page 16-19, line 389-406). Two figures lack scale for measurement in Figure 5 (Page 13, line 345-346).
  6. Lacks of most statistics analysis in Figure 15 (Page 24, line 461-470). Please detailed compare grouped data in every tissues. Now only tumor tissue has been analyzed.

Author Response

Reviewer 3 comments:

  1. In the part of “Cytotoxicity of CA-in-DMEM, CA-in-DMB and HA” controls of other breast cancer cell lines and several normal cells are needed.

Answer: Addressed in Reviewer 1: comment 1

  1. It's better to be able to verify the toxicity effect on mice.

Answer: We appreciate the suggestions. To perform these experiments, we need to acquire animal ethics approval which usually takes several months.

  1. In this paper, authors employed subcutaneous tumor model. However, to be more convincing, orthotopic breast cancer model should also be implemented.

Answer: It was a mistake made while reporting. Our model is an orthotopic breast cancer model developed by the method described in the video article by Paschall and Liu [1]. We corrected it.

Line 283-285: “Approximately 5 × 105 4T1 cells in 100 μl PBS were injected subcutaneously in the mammary pad of mice (considered as day 1) and the mice were checked regularly for the outgrowth of tumour by touching the area of injection.” has been changed to “5 × 105 4T1 cells in 100 μl PBS were injected in the mammary fat pad of a mouse to induce an orthotopic breast tumour (considered as day 1) and the mice were checked regularly for the outgrowth of tumour by touching the area of injection.”

  1. And should clearly clarify the data directly support the result “tumors were approximately 100 mm3 in volume”.

Answer: We checked our raw data and corrected this to exact value.  Line 286: “approximately 100 mm3 in volume” has been corrected to “105 ± 8 mm3 in volume”.

  1. Why only used 4 hours treatment? Better set a series of time points, to observe the accurate dynamic biodistribution of NPs in the organs.

Answer: We appreciate the suggestions. To perform these experiments, we need to acquire animal ethics approval which usually takes several months.

  1. Better at least six mice were used to animal test.

Answer: We agree that the more the better. We can increase the sample size for our future studies. However, it is not practical to repeat the assay with higher number of animals at this point of time.

Also, n = 5 is quite acceptable in research community (ResearchGate). As for reference of acknowledged papers, Ya Liu et al. reported n = 4 for the biodistribution study of polymeric amphiplillic NP in carps [2], Leila Hasanzadeh et al. reported n = 3 for their biodistribution study of CeO2 NP in mice [3].

  1. “The of size of ” should be corrected to “The size of ” (Page 1, line 39)

Answer: Corrected.

  1. In the material and method part, the information of where 4T1 cells are purchased from, and how to culture, is missing.

Answer: Line 136: Information about 4T1 purchase is added.

Line 206, 211: Cell culture method is updated.

  1. Description of statistics analysis is not clear enough.

Answer: Section 2.12. Statistical analysis has been updated. Line 298-302:

For MTT assay, two sample unpaired t-test was performed to analyse the difference between untreated and NP treated groups. Correlation coefficient was calculated for particle size with increasing calcium concentration. Two sample paired t-test was performed for stability test. For in vivo biodistribution study, single factor ANOVA was run to compare the groups. Data was considered statistically significant at p value <0.05.

  1. The rulers in Figure 2 , 3 and 13 is not clear, and one ruler in Figure 13 is not placed in the bottom right-hand corner(Page 10-11, line 326-332).

Answer: Corrected. Also, scale added in figure 3.

  1. Lack of statistics analysis in Figure 4, 8-11 (Page 12,line 338-339), (Page 15-16, line 373-378), (Page 16-19, line 389-406). Two figures lack scale for measurement in Figure 5 (Page 13, line 345-346).

Answer: Figure 4: Correlation coefficient calculated and described in lines 345-347.

Figure 8: Statistical analysis added on figure. Line 387: “ ‘*’ indicates significant difference at p < 0.05.” added at figure caption.”Lines 377-381: Text changed from “NPs-siRNA complex for these two particles were stable even after a rigorous wash with pure water, which was evident from Figure 8B, with a slight reduction for HA with 10 nM of siRNA. In contrast, siRNA binding to CA-in DMEM was concentration-dependent and better with lower concentration of siRNA (Figure 8A).” to “NPs-siRNA complex for these two particles were stable for lower concentration of siRNA even after a rigorous wash with pure water. However, binding efficiency reduced significantly for 10 mM of siRNA which was evident from Figure 8B. In contrast, siRNA binding to CA-in DMEM was concentration-dependent and significantly reduced with increasing siRNA concentration (Figure 8A).

Figure 9: redrawn with standard deviation. Added in lines 410-412: Two sample paired T-test was performed for statistical analysis. For A,B, C, compared to day-0, all data were significant. For pre-prepared CA, compared to day-0 data, particle formation was significantly higher when stored at -20 °C and -80 °C (E, F).

Lines 393-406: “CA NPs prepared with DMEM stored at three different temperatures- +4 °C, -20 °C and -80 °C, gave higher absorbance than freshly prepared NPs. CA prepared with DMEM stored at +4 °C gave similar absorbance after three time points- 1 day, 7 days and 30 days, whereas the absorbance increased with time for DMEM stored at -20 °C and -80 °C (Figure 9A, 9B, 9C). The pre-prepared CA stored at -20 °C and -80 °C gave high absorbance from 1 mM Ca2+. On the contrary, particles stored at +4 °C seemed to be degraded with time (Figure 9D, 9E, 9F).”has been changed to “CA NPs prepared with DMEM stored at three different temperatures- +4 °C, -20 °C and -80 °C, gave higher absorbance than that for freshly prepared NPs (Day 0). CA prepared with DMEM stored at +4 °C gave similar absorbance after three time points (Fig 9A). However, while compared to day-0 data, day-1 and day-7 data were significant at p < 0.05, data for day-30 were significant at p<0.1 and seems to produce less particles at low Ca2+ concentration. The absorbance increased with time for DMEM stored at -20 °C and -80 °C (Figure 9B and 9C). While the chart shows a pattern of higher absorbance indicating more particles, data were statistically significant at p <0.5. with an exception for day 0 versus day 30 data for DMEM stored at -80 °C which was very significant at p < 0.01.

The pre-prepared CA stored at stored at +4 °C seemed to be degraded with time (Figure 9D), with significantly low absorbance for day-30 data. Pre-prepared CA stored at -20 °C and -80 °C gave significantly high absorbance from 1 mM Ca2+compared to that for NP prepared with fresh buffer with statistical significance at p < 0.1 (Figure 9E, 9F). In a word, we can conclude that CA NP stored at +4 °C degrades with time. However, colder temperature supports increased particle formation.”

 Figure 10: redrawn with standard deviation. Added in lines 420-421:  Two sample paired T-test was performed for statistical analysis.

Lines 416-418: “The buffer started to lose its integrity by 24 hours which was evident from the absorbance of CA prepared using the stored buffer.” Changed to “The buffer started to lose its integrity by 24 hours which was evident from the reduction in absorbance values of CA prepared using the stored buffer. Compared to day-0, absorbance was significantly low on day 30 at p < 0.1 for DMB stored at -80 °C.

Figure 11: redrawn with standard deviation. Added in lines: 431-433: Two sample paired T-test was performed for statistical analysis. Compared to freshly prepared HA, significantly more particles were formed with HBS stored at 4 °C for 30 days and at -80 °C for 7 and 30 days.

Lines 424-429: “HA prepared with HBS stored at 4 °C and-20 °C for 1, 7 and 30 days gave absorbance values, similar to that of freshly prepared particles. Absorbance values increased initially and then gradually decreased for HBS stored at -80 °C (Figure 11).” Changed to “HA prepared with HBS stored at 4 °C and-20 °C gave absorbance values, similar to that of freshly prepared particles (Figure 11A and 11B) and were not significantly different, except for day-30 data for HBS stored at +4 °C which was very significantly higher than day-0 data at p < 0.001. For HBS stored at -80 °C (Figure 11C), particle formation was significantly higher on day 7 and day 30, compared to that for fresh HBS on day 0 (p < 0.01). We can consider that HBS can maintain its integrity at 4 °C for a duration smaller than 30 days and at -20 °C for 30 days.

Figure 5: measurements has been added manually as the image was zoomed out. Page 12-13.

  1. Lacks of most statistics analysis in Figure 15 (Page 24, line 461-470). Please detailed compare grouped data in every tissues. Now only tumor tissue has been analyzed.

Answer: Line 485: “Signal was very poor’ was changed to “signal was negative”.

Added in lines 488-490: Among the NPs and free siRNA there was no statistically significant difference in their levels in spleen, heart, lung and liver.

Added in lines 500-501: No NP was detected in kidneys and brain. CA-in-DMEM was not detected in heart and lungs. Spleen and liver showed presence of free siRNA as well as NPs. However, there was no significant difference.

Turnitin report:

We didn’t find plagiarism.

Round 2

Reviewer 3 Report

The manuscript is much improved after the revision. In order to further verify the results in the paper, it is suggested more experiments in other cell and animal models after the article was published.